

# Disinfection effect of povidone-iodine in aquaculture water of swamp eel (*Monopterus albus*)

Xuan Chen[1,*], Chongde Lai[1,*], Yulan Wang[2], Lili Wei[3] and Qiwang Zhong[1]

[1] College of Biological Science and Engineering, Jiangxi Agricultural University, Nanchang, China
[2] Nanchang Academy of Agricultural Sciences, Nanchang, China
[3] College of Animal Science and Technology, Jiangxi Agricultural University, Nanchang, China
[*] These authors contributed equally to this work.

## ABSTRACT

The swamp eel (*Monopterus albus*) is an important commercial farmed fish species in China. However, it is susceptible to *Aeromonas hydrophila* infections, resulting in high mortality and considerable economic loss. Povidone-iodine (PVP-I) is a widely used chemical disinfectant in aquaculture, which can decrease the occurrence of diseases and improve the survival. However, environmental organic matter could affect the bactericidal effectiveness of PVP-I, and the efficacy of PVP-I in aquaculture water is still unknown. In this paper, disinfection assays were conducted to evaluate the effectiveness of PVP-I against the *A. hydrophila* in different types of water. We found that the effective germicidal concentration of PVP-I in outdoor aquaculture water was 25 ppm for 12 h. In indoor aquaculture water with $10^5$ CFU/mL bacteria, 10 ppm and 20 ppm of PVP-I could kill 99% and 100% of the bacteria, respectively. The minimal germicidal concentration of PVP-I in Luria-Bertani broth was 4,000 ppm. Available iodine content assay in LB solutions confirmed that the organic substance had negative impact on the effectiveness of PVP-I, which was consistent with the different efficacy of PVP-I in different water samples. Acute toxicity tests showed that the 24 h-$LC_{50}$ of PVP-I to swamp eel was 173.82 ppm, which was much higher than the germicidal concentrations in outdoor and indoor aquaculture water, indicating its safety and effectivity to control the *A. hydrophila*. The results indicated PVP-I can be helpful for preventing the transmission of *A. hydrophila* in swamp eel aquaculture.

## INTRODUCTION

The swamp eel (*Monopterus albus*), belongs to Order Synbranchiformes, Family Synbranchidae, and is widely distributed in southern China, Japan, India and other Southeast Asian countries (FishBase, http://fishbase.org/). Due to its great growth performance and rich nutrient content, swamp eel has become a commercially important farmed species in China. However, the intensive and stressful rearing conditions make farmed swamp eels highly susceptible to bacterial pathogens, such as *Aeromonas hydrophila*, which is the main pathogen causing hemorrhagic septicemia and result in high mortality

Corresponding authors
Lili Wei, hbliliwei@163.com
Qiwang Zhong,
zhongqiwang@jxau.edu.cn,
zhongqw2000@163.com

and considerable economic losses (*Hossain et al., 2014*; *Nielsen et al., 2001*). The clinical signs of infection include red and swollen anus, visceral congestion, skin erythema and gill hemorrhage (*Jagoda et al., 2014*).

The hemorrhagic septicemia severely restricts the development prospects of the swamp eel farming industry. To prevent the spread of *A. hydrophila* and outbreak of the diseases in aquaculture, the direct and effective method is to reduce the amount of pathogenic bacteria in the aquaculture water. Povidone-iodine (PVP-I) is an important chemical disinfection widely used to disinfect pathogenic organisms and equipments in aquaculture (*Scarfe, Lee & O'Bryen, 2006*). It has been reported as a broad-spectrum microbicide with potency to inactivate bacteria, fungi, protozoans, several viruses and some spores (*Wutzler et al., 2000*). Moreover, PVP-I was determined to be an animal drug of low regulatory priority by the Food and Drug Administration, and suggested to act as an egg surface disinfectant due to the lower irritation and toxicity to tissues (*USFDA, 2010*). However, many environmental factors can affect the efficacy of PVP-I, such as temperature, pH, organic matter, etc. (*Amend, 1974*). The presence of organic matter can result in a significant decrease in the bactericidal effectiveness of PVP-I (*Rodriguez Ferri et al., 2010*). By far, most disinfection research has been carried out in sterile water (*Chang et al., 2015*), and only very few reports mention use of aquaculture water for testing (*Hershberger, Pacheco & Gregg, 2008*).

Swamp eel aquaculture includes outdoor cage culture and indoor tank culture. The former is mainly used for large-scale commercial swamp eel production, and the latter is commonly used in laboratory research. Outdoor aquaculture ponds are complex environments containing large amounts of organic matter and suspended solids (*Lin, 2002*). However, indoor tank culture mainly uses dechlorinated tap water with water changes every day, so the content of organic matter of indoor culture water is very low. Different contents of organic matter may cause differences in disinfection efficiency of PVP-I (*Yoneyama et al., 2006*). Accordingly, it is necessary to test whether the organic matter in the aquaculture water is able to cause significant negative effect on disinfection.

The aim of this study was to investigate the germicidal effect of PVP-I on *A. hydrophila* in three different solutions and attempt to determine the effective disinfection concentrations to control the transmission of *A. hydrophila*.

## METRIALS AND METHODS

### Bacterial strain

Six isolates of *A. hydrophila* used in this study (Ah 1-6) were isolated from sick *M. albus* by our laboratory and stored at −80 °C until used. Before disinfection assays, all isolates were subcultured at 25 °C overnight on solid-phase Luria-Bertani (LB) agar (10 g/L tryptone, 5 g/L yeast extract, 10 g/L sodium chloride and 15 g/L agar powder).

### Water sample preparation

Three different water samples were prepared in this study: outdoor net cage water, dechlorinated tap water and LB broth. These three types of water samples represented outdoor aquaculture waters, indoor aquaculture waters and eutrophic waters, respectively. Outdoor aquaculture waters were sampled from four different net cages of a swamp eel

farm, and some of the swamp eel in one cage had typical clinical signs of hemorrhagic septicemia. Tap water samples were filtered through 0.22 $\mu$m membrane to remove impurities and microbes and autoclaved at 121 °C for 15 min. LB broth was prepared with deionized distilled water.

## Germicidal test

PVP-I solutions were prepared with different water samples immediately prior to use. For outdoor aquaculture water, 1 mL of each net cage samples were treated with 5 to 100 ppm PVP-I (final concentration) at 25 °C in the dark for different durations (0, 2, 4, 8 and 12 h).

For indoor aquaculture water (tap water) and LB solution, *A. hydrophila* strains were used to evaluate the disinfection effect of PVP-I. Briefly, for indoor aquaculture water, six *A. hydrophila* isolates (Ah 1-6) were incubated in LB liquid media overnight at 25 °C, respectively. Then 5 mL of each culture broth was centrifuged at 12,000 rpm for 5 min at 25 °C. The bacterial pellet was washed with tap water three times and then resuspended and diluted to different concentration ($10^3$–$10^5$ CFU/mL). Disinfection treatments were performed by mixing 900 $\mu$L of diluted bacteria solutions and 100 $\mu$L PVP-I solution to a 1 mL volume containing $10^3$–$10^5$ CFU/mL bacteria and 1.25–20 ppm of PVP-I (final concentration). For LB liquid media, only *A. hydrophila* Ah1 isolate (isolated from skin) was used, and the disinfection treatments were performed like those for the indoor aquaculture water except that the tap water was replaced by LB liquid media.

For all above disinfection trials, treatment tubes were incubated at 25 °C in the dark with gentle mixing. At each nominal exposure time (0, 2, 4, 8 and 12 h), a 160 $\mu$L aliquot was transferred to a sterile 1.5 mL microcentrifuge tube, and the equal volume of sodium thiosulfate (0.004 mol/L) was added to neutralize the PVP-I immediately. After neutralization, the mixed solution was tenfold serially diluted, and then 100 $\mu$L of each diluted solution was plated onto LB agar plates in triplicate using sterile beads (*Sanders, 2012*). Plates were incubated at 25 °C for 24 h and colonies were counted. Every experiment was repeated three times.

## Effect of organic matter on available iodine

LB liquid media was 10-fold serially diluted with ddH$_2$O. Then the 2,000 ppm of PVP-I solutions were prepared with serially diluted LB solutions. 1 mL of PVP-I solution was mixed with 2 mL of 0.1% soluble starch solution, and the absorption was measured at 585 nm immediately. The undiluted LB broth and ddH$_2$O were used as controls. The experiment was repeated three times.

## Median lethal concentration (LC$_{50}$)

Healthy swamp eel (11–15 g) were obtained from a commercial farm and acclimated in dechlorinated tap water at 25 $\pm$ 1 °C in 10 L aquarium tanks for 2 weeks until use. Daily, eels were fed and the tank water replaced with fresh water. The fishes were starved for 24 h prior to and during the test to reduce the contaminations by fecal and excess food. Five concentrations of PVP-I were chosen for testing purposes (100, 150, 175, 200 and 250 ppm) and a group without PVP-I was used as the control. The duration of exposure was 24 h. For each concentration and control, three replications were conducted and each

replication contained ten swamp eel. Fish were observed every four hours and the dead fish was immediately removed from the test tanks. The $LC_{50}$ value was calculated using the Probit analysis (*Lin, 2002*; *Lu et al., 2017*). After the $LC_{50}$ test, the surviving individuals were replaced in dechlorinated tap water without PVP-I, fed and the tank water replaced with fresh water daily. The status and survival of the eels were monitored for at least one month. All animal procedures were conducted according to the guidelines for the care and use of experimental animals established by the Ministry of Agriculture of China (No. SCXK YU2005-0001). Animal Care and Use Committee (ACUC) in Jiangxi Agricultural University specially approved this study.

### Statistical analysis

SPSS17.0 was used for data analysis. Significance was evaluated by one-way analysis of variance (ANOVA) using LSD test. A value of $P < 0.05$ was considered to indicate a significant difference. Probit analysis was used to calculate the median lethal dose.

## RESULTS

### The effect of PVP-I in outdoor aquaculture water

Four outdoor net cage water samples were marked as WA, WB, WC and WD. pH value of these cage water samples were 7.12, 7.25, 7.08 and 7.32, respectively. The water sample WD was taken from a cage in which hemorrhagic septicemia was discovered. According to plate count, the initial cultivable bacterial concentration of these samples were $(0.70 \pm 0.07) \times 10^3$ CFU/mL, $(1.59 \pm 0.23) \times 10^3$ CFU/mL, $(0.38 \pm 0.04) \times 10^3$ CFU/mL and $(3.96 \pm 0.22) \times 10^3$ CFU/mL, respectively. The germicidal test showed that low concentrations of PVP-I, such as 5 ppm and 10 ppm, could not provide effective disinfection (Fig. 1). When treated with these two concentrations of PVP-I, the average survival decreased in early period, then increased and finally up to about 1.5–2.5 fold and 0.8–1.5 fold of initial bacteria amount, respectively. Increasing the PVP-I concentration to 25 ppm significantly improved the bactericidal effects resulting in $98.08 \pm 0.17\%$, $98.74 \pm 0.13\%$, $96.49 \pm 0.03\%$ and $99.66 \pm 0.03\%$ mortality in four outdoor water samples for 12 h, respectively (Fig. 1). However, higher PVP-I concentrations (50–100 ppm) did not significantly increase the sterilization rate ($P > 0.05$), although 80 ppm and 100 ppm could kill all bacteria in WA and WD samples.

### The effect of PVP-I in indoor aquaculture water

In small-scale indoor farming, the dechlorinated tap water is often used as aquaculture water. The results showed that the effects of PVP-I on six *A. hydrophila* isolates was similar in tap water, and with the increase of bacterial content, the concentration of PVP-I for complete sterilization increased accordingly. In tap water with $10^3$ CFU/mL bacteria (Fig. 2A), 1.25 ppm of PVP-I (final concentration) could reduce bacterial counts within 12 h, but did not completely eliminate the *A. hydrophila*. Increasing the concentration of PVP-I to 2.5 ppm, all bacteria were killed within 4 h. When the bacterial content was $10^4$ CFU/mL (Fig. 2B), 10 ppm of PVP-I was required to achieve complete disinfection within 2 h, and lower concentrations (2.5 ppm and 5 ppm) could not kill all bacteria.

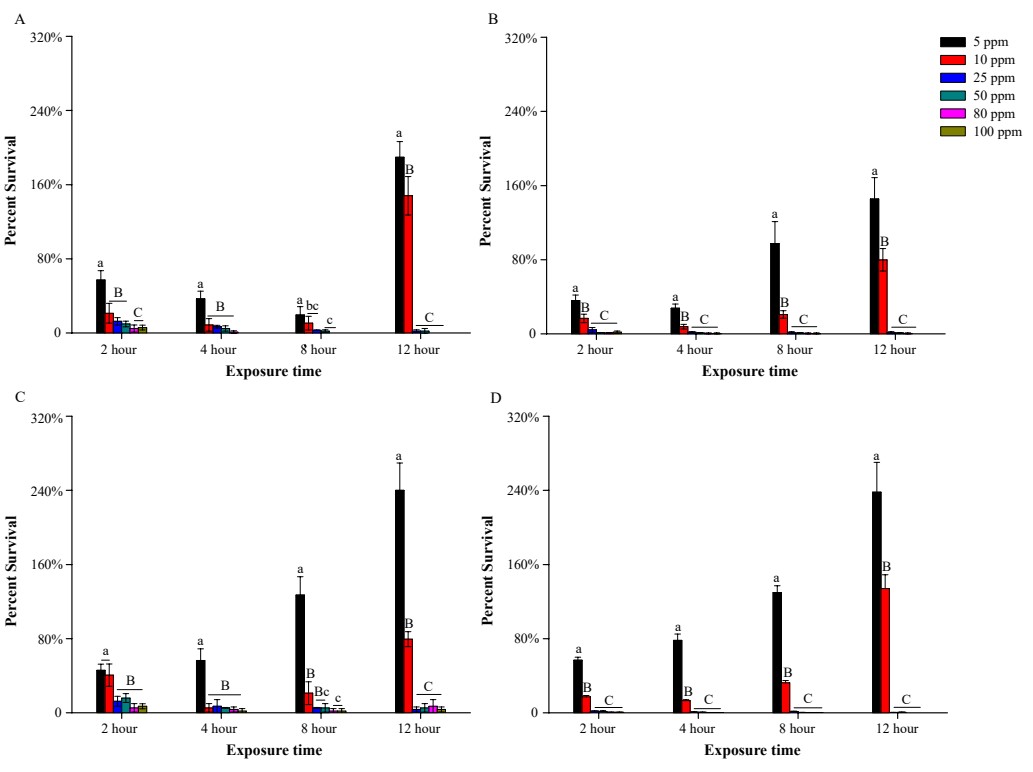

**Figure 1  Bactericidal efficacy of different povidone-iodine concentrations in four outdoor aquaculture water WA (A), WB (B), WC (C) and WD (D).** At each time point, the capital letters above the columns show extremely significant differences ($P < 0.01$); the little letters above the columns show significant differences ($P < 0.05$). The $y$-axis indicates the percent survival [percent survival = (CFU of treatments/CFU of controls) ×100%]. Data are shown as means ± SEM ($n = 3$).

When the bacterial content was $10^5$ CFU/mL (Fig. 2C), although 10 ppm of PVP-I could kill more than 99% of the bacteria, to achieve 100% of sterilization 20 ppm of PVP-I was required. When treated with lower concentrations, the amount of culturable bacteria gradually declined to the lowest in 8 h, and then increased to different levels.

## The effect of PVP-I in LB liquid media

LB liquid media was used as eutrophic controls. In LB liquid media, neither 500 ppm nor 1,000 ppm of PVP-I could inhibit the proliferation of *A. hydrophila* at all bacterial concentrations. For example, in LB solution with $10^3$, $10^4$ and $10^5$ CFU/mL bacteria (Fig. 3), after 12 h of incubation with 500 ppm of PVP-I, the number of bacteria increased to 1,800, 6,050 and 20,000 times to the initial number of bacteria, respectively. Increasing the PVP-I concentration to 2,000 ppm only controlled bacteria at the lowest bacteria concentration ($10^3$ CFU/mL of LB medium; Fig. 3A). To achieve complete disinfection, 4,000 ppm were required and in this concentration all *A. hydrophila* could be killed within 2 h.

## Available iodine measurement

LB medium contained a large amount of organic matter, so it was used to evaluate the effect of organic matter on the available iodine content in this study. The results showed that the

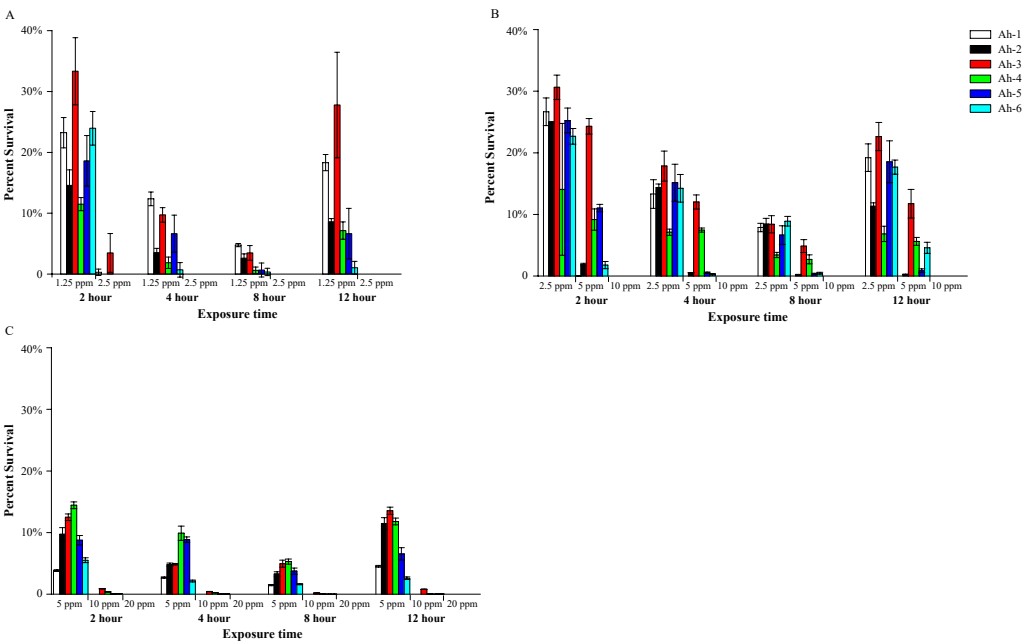

**Figure 2  Bactericidal efficacy of povidone-iodine against six Aeromonas hydrophila isolates (Ah 1-6) in indoor aquaculture water (tap water).** The test bacterial concentration were $10^3$ CFU/mL (A), $10^4$ CFU/mL (B) and $10^5$ CFU/mL (C). The $y$-axis indicates the percent survival. Data are shown as means $\pm$ SEM ($n = 3$).

concentration of organic matter could significantly affect the available iodine content, and the higher LB concentrations lead to the lower available iodine contents (Fig. 4). In 2 g/L PVP-I solutions, when the concentration of LB was less than 1%, the content of available iodine was equivalent with that in ddH$_2$O. With the increase of LB concentration, the available iodine content decreased rapidly, and when the LB concentration was 20%, the effective iodine content almost reduced to zero.

## LC$_{50}$ test

Median lethal dose of PVP-I was calculated using the Probit analysis (Table S1). According to the equation: $\mathrm{Probit(p)} = -5.214 + 0.03 \times \lg(\mathrm{dose}) (\chi^2 = 6.343)$, the 24h-LC$_{50}$ of PVP-I to swamp eel was 173.82 ppm. The individuals that survived the LC$_{50}$ test did not die within the month following exposure.

## DISCUSSION

Disease is a main threat in aquaculture production and disinfection of water bodies has been used as an important measure to prevent waterborne pathogen transmission in aquaculture (*Scarfe, Lee & O'Bryen, 2006*). The present disinfection tests were mainly carried out in sterile water (*Hershberger, Pacheco & Gregg, 2008*; *Mainous, Smith & Kuhn, 2010*), while very few in aquaculture water. However, there is a great differences between sterile water and aquaculture water, and the results obtained in sterile water are not suitable

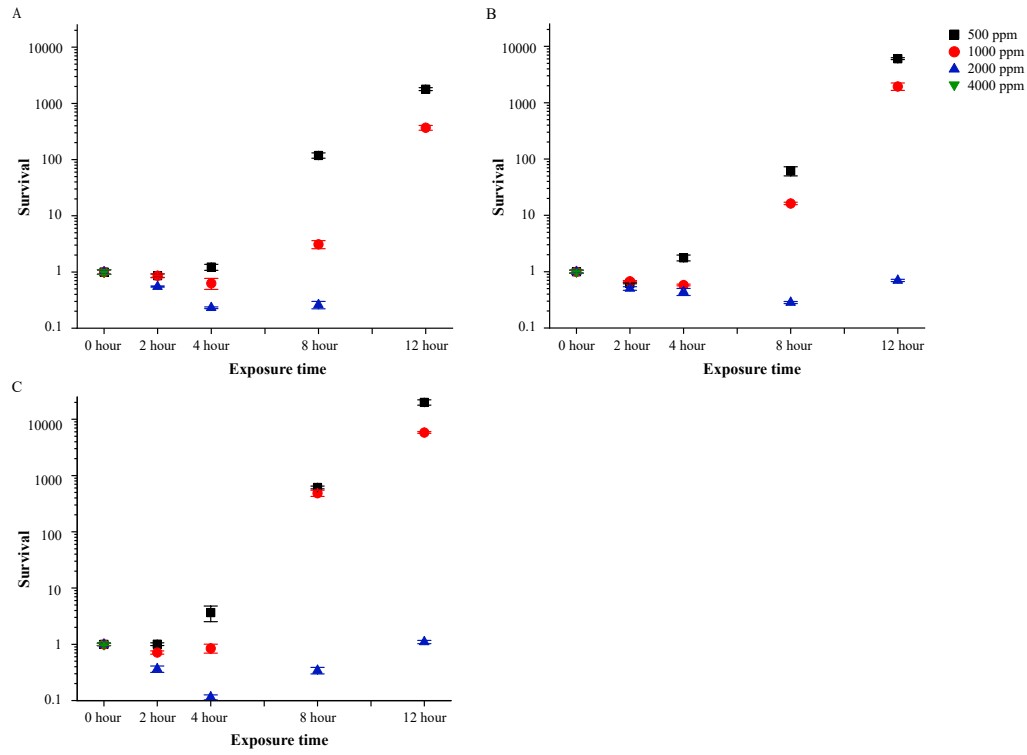

**Figure 3  Bactericidal efficacy of povidone-iodine against *Aeromonas hydrophila* isolate (Ah 1) in Luria-Bertani broth. The tested bacterial concentration were $10^3$ CFU/mL (A), $10^4$ CFU/mL (B) and $10^5$ CFU/mL (C).** The $y$-axis indicates the non-percent survival (non-percent survival = CFU of treatments/CFU of controls). Data are shown as means ± SEM ($n = 3$).

for aquaculture water. The aim of this study was to evaluate the effective concentration of PVP-I against *A. hydrophila* in outdoor aquaculture water and indoor aquaculture water.

Our study showed that PVP-I was effective to prevent *A. hydrophila* proliferation in outdoor and indoor aquaculture water and there were water-specific differences in susceptibility. The effective germicidal concentration of PVP-I in outdoor aquaculture water, indoor aquaculture water (tap water) and eutrophic water (LB broth) were 25 ppm, 10 ppm and 4,000 ppm, respectively. These water-specific differences in bactericidal effect might be mainly caused by organic matter. It had been proved by serially diluted LB solution, in which the higher ratio of organic matter (LB medium) lead to the lower concentration of available iodine, and 20% LB was sufficient to neutralize all the free iodine in 2,000 ppm PVP-I solutions. This result was also supported by many reports which indicated that the efficacy of PVP-I declined when organic matter (e.g., blood, fish mucus, amino acids, or simple aromatic compounds) was present (*Truesdale & Luther, 1995*; *Yoneyama et al., 2006*). Similar results also had been observed in seawater that the content of molecular iodine reduced within hours when added to seawater (*Truesdale, Luther & Canosa-Masb, 1995*). Considering the differences in different aquaculture water, it was necessary to determine the effective disinfectant concentration before use. It should be also noted that, in addition to the effects of a small amount of organic matter in tap

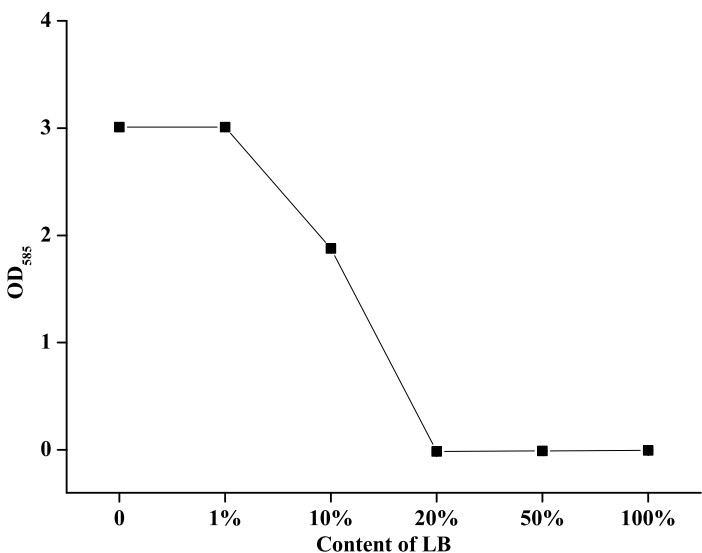

**Figure 4** **Effect of different LB concentrations on the effective iodine content of povidone-iodine.** Data are shown as means $\pm$ SEM ($n = 3$).

water, the inorganic matter, such as the hardness of the water, had been proved to affect the available iodine content by influencing pH (*Amend, 1974*).

Disinfection can be used not only as a preventive measure in ponds or for equipments, but also as a remedial measure after the outbreak of the disease. The majority of commercially available disinfectants are very effective when used at high concentrations within short contact time. However, for fish, prolonged exposure to high concentrations of disinfectants might be dangerous (*Bergmann, Monro & Kempter, 2017*). Especially for extensive outdoor aquaculture, it is impossible to change the water after disinfection. Therefore, it was necessary to determine the toxicity of disinfectants to fish (*LeValley, 1982*). In this study, the 24 h-LC$_{50}$ for *M. albus* was 173.82 ppm, which was much higher than the effective germicidal concentration in aquaculture water, suggesting that PVP-I could be used safely as a disinfectant for the culture of *M. albus*. On the other hand, the commercial standard procedure for PVP-I is to pour directly into the water with the final concentration of 0.075–0.093 ppm for prevention and 0.093–0.125 ppm for treatment (*Wang et al., 2015*). Our data showed that these recommended concentrations were far less than the effective concentrations and would not have any germicidal effect on *A. hydrophila*.

Compared with higher PVP-I concentration, the lower PVP-I concentrations reduced the toxicity to *M. albus*, but reduced the bactericidal effect. For example, in outdoor aquaculture water, indoor aquaculture water (tap water) and LB medium, when treated with non-lethal concentration, the number of cultivable bacteria decreased first, then increased during 12 h. Similar phenomenon was also reported in seawater when treated with some disinfectants (sodium hypochlorite, bleaching powder, formalin) for 12 h (*Wang et al., 2015*). The increase of bacteria in later period might be due to the decrease of available iodine content and the proliferation of surviving bacteria. On the one hand,

the content of available iodine decreased significantly with the passage of time (*Chang et al., 2015*); on the other hand, the organic matter in the solution would neutralize free iodine (*Takeda et al., 2016*). Meanwhile, the organic matter in water samples could supply nutrients for the proliferation of the surviving culturable bacteria. Our results supported this hypothesis; the earliest rise and largest increase of the number of *A. hydrophila* happened in LB broth which contains the most abundant organic matter, followed by outdoor aquaculture water and tap water. Moreover, *Kersters et al. (1996)* reported that *A. hydrophila* was able to grow and proliferation in nutrient-poor filtered and autoclaved tap water, which made it hard to control the proliferation of *A. hydrophila* effectively.

It should be noted that we did not consider the biofilm formation and viable but non-culturable (VBNC) state of *A. hydrophila*. It was reported that the biofilms of *A. hydrophila* was more resistant to disinfectants than planktonic cells (*Jahid & Ha, 2014*). Although there was no work about the relationship between the VBNC of *A. hydrophila* and disinfection, it has been demonstrated that stressed and starved *A. hydrophila* could enter a VBNC state (*Pianetti et al., 2008*; *Rahman, Suzuki & Kawai, 2001*). Disinfectants, such as hypochlorous acid, similarly have induced *Escherichia coli* and *Salmonella typhimurium* into the VBNC state (*Oliver, Dagher & Linden, 2005*). In this study, when PVP-I concentrations were lower than the effective sterilization concentration, the bacteria might not be completely killed and some of them went into the VBNC state. So the early decline in the number of bacteria might be caused by death cells and VBNC state cells together. As for the subsequent increase in the number of bacteria in outdoor aquaculture water, tap water and LB medium, whether it was associated with the resuscitation and growth of some mildly injured VBNC cells, further study was needed. But, there was some evidence that VBNC bacteria could resuscitate and proliferate under certain conditions (*Dukan, Levi & Touati, 1997*). Moreover, the use of disinfectant might result in the development of resistant strains. Nuñez reported the resistant gram-negative rods strains for chlorhexidine, such as *A. hydrophila*, *Shigella flexneri* (*Nuñez & Moretto, 2007*). In addition, resistant *Staphylococcus* sp. for PVPI were also isolated from hospital wastewater (*Nuñez & Moretto, 2007*) and some instances have been described that iodophors had been found to be contaminated with *Pseudomonas* sp (*Anderson et al., 1984*). However, there has been no report of *A. hydrophila* tolerant to PVPI so far.

This study confirmed the effect of organic matter on PVP-I sterilization and suggested that aquatic organic matter should be considered when PVP-I was used in aquaculture. The results showed the great bactericidal activities of the PVP-I in different aquaculture waters, and we recommended 25 ppm for outdoor aquaculture and 20 ppm for indoor treatment, which were safe and effective for normal swamp eel farming.

## CONCLUSIONS

In conclusion, the organic matter in aquaculture water has negative influence on the bactericidal effectiveness of PVP-I. The minimum effective bactericidal concentration of PVP-I in indoor aquaculture water and outdoor aquaculture water was 10 ppm and 20

ppm, respectively. The results suggest PVP-I could help prevent the transmission of *A. hydrophila* in swamp eel aquaculture.

### Funding

This work was supported by grants from the National Natural Science Foundation of China (No. 31160530 and No. 31360634), the Science and Technology Support Program of Jiangxi Province (20122BBF60074), the Natural Science Foundation of Jiangxi Province (20114BAB214003) and the Project of Education Department in Jiangxi Province (GJJ13289). The funders had no role in study design, data collection and analysis, decision to publish, or preparation of the manuscript.

### Grant Disclosures

The following grant information was disclosed by the authors:
National Natural Science Foundation of China: 31160530, 31360634.
Science and Technology Support Program of Jiangxi Province: 20122BBF60074.
Project of Education Department in Jiangxi Province: GJJ13289.
Natural Science Foundation of Jiangxi Province: 20114BAB214003.

### Competing Interests

The authors declare there are no competing interests.

### Author Contributions

- Xuan Chen performed the experiments, analyzed the data, prepared figures and/or tables, approved the final draft.
- Chongde Lai and Yulan Wang contributed reagents/materials/analysis tools.
- Lili Wei contributed reagents/materials/analysis tools, authored or reviewed drafts of the paper.
- Qiwang Zhong conceived and designed the experiments, authored or reviewed drafts of the paper, approved the final draft.

### Animal Ethics

The following information was supplied relating to ethical approvals (i.e., approving body and any reference numbers):

All experimental procedures were carried out in accordance with the guidelines in the China Law for Animal Health Protection and Instructions (Ethics approval No. SCXK (YU2005-0001).

### Data Availability

The raw data are provided as Supplemental Information 1.

## Supplemental Information

Supplemental information for this article can be found online at http://dx.doi.org/10.7717/peerj.5523#supplemental-information.

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
