# Peer review of "Disinfection effect of povidone-iodine in aquaculture water of swamp eel (Monopterus albus)"

_PeerJ, doi:10.7717/peerj.5523_

## Round 0.1 · original submission · Major Revisions

The reviewers have commented on your paper. They indicated that it is not acceptable for publication in its present form. I suggest you modify your manuscript taken into account all the suggestions given by reviewers. Special attention should be paid to those suggestions related to the experimental design.

Reviewer 1 ·

Basic reporting

Please clarify what the authors mean by 'Line 229: After the test, none of the survivals was dead within a month.'
Was the dose of 173.82 ppm lethal to the eels?

Please work on improving sentence construction in some places.
Example: Discussion: Disease is a main threat in aquaculture production and disinfection of water bodies has been used as an important measure to prevent waterborne pathogen transmission in aquaculture' and 'However, there is a great differences between sterile water and aquaculture water, and this might lead to wrong guidance in practice.

Experimental design

It is widely known that presence of organic matter will decrease the efficiency of antimicrobials such as iodine. I did not see a point in testing iodine efficiency with sterile and distilled water.
The authors should instead add/include results for water testing from various aquaculture farms and determine the amount of iodine necessary to control Aeromonas hydrophila in real conditions.

Validity of the findings

The findings are not entirely novel as it is known that organic matter will decrease availability of disinfectants like iodine. Results from actual farm water samples or if not available, simulating farm water conditions and testing them would make the study more robust.

·

Basic reporting

There are several sentences that require attention in terms of English and spellings.
Line 33: ...wildly --> ...widely
Line 61-63: It is better to use reliable reference/s to demonstrate the distribution of this swamp eel, such as the FishBase
Line 81: ...factor --> ...factors
Line 82: ..., PH,... --> ..., pH,...
Line 99: ... the pathogen... --> ... A. hydrophila... (italic)
Line 168: ...these were... --> ... these cage water samples were ...
Line 323: ... water had ... --> ... water has ...
Line 323-324: ...b actericidal ... --> ... bactericidal ...
Line 326: ... A. hydrophila ... --> ... A. hydrophila ... (italic)
Line 357: ... Health. 29... --> ... Health 29 ...
Line 396-397: Title of article should be in sentence form, not using Capital alphabet.

Figure 1: Label properly the (A), (B), (C) and (D) in the figure.

Experimental design

Research questions are well defined.
The experiment was designed properly. There is no issue about animal ethics as this study is following guidelines by the ministry and have been approved by ACUC of the university.
The methods described are sufficient.

Validity of the findings

Results are discussed in a good manner.
Conclusions are written well to answer the research objectives.

---

## Round 0.2 · Major Revisions

The two prior reviewers were not available to re-review, therefore I had to invite a third reviewer.

As you can see, this reviewer has identified some important deficiencies which should be solve in the revised version. Please, take into account all the suggestions given in order to improve the text.

·

Basic reporting

The article needed some editing to improve the English, but the writing was better than I have seen in many articles written by authors for whom English is not their native language. See attached word file of the document for specific suggestions for rewording. The introduction adequately introduced the work conducted. The discussion could be improved by broader discussion of the iodine literature, disinfection in eel culture, and microbial resistance.
Figure font size needs to be increased to make it more readable.

Experimental design

see general comments for a discussion of the controls used in the study.

Validity of the findings

The results contribute some locally useful information. However the effects of organic matter have already been reported and general bactericidal effects of iodine are already well known.

Additional comments

‘Disinfection effect of povidone-iodine in aquaculture water of swamp eel (Monopterus albus)’
Review: The manuscript summarizes research on the effects of povidone iodine on water containing isolates of A. hydrophila. The paper provides some useful data on the concentrations of iodine needed to kill bacteria and on the effect of organic matter on effective concentrations within an eel culture context. There are several deficiencies in the research, but there is enough good data that I recommend publishing after some revision. I have made some grammatical suggestions in the attached word file using track changes. Other suggestions for improving the manuscript and concerns to address are as follows:
1. The study inappropriately considers the deionized water treatment and LB broth treatment (high organic load) as negative and positive controls. These are additional treatment variables not controls. Proper controls for the Germicidal Test would be water samples with (positive control) and without (negative control) A. hydrophila to which no chemical was added. The data for the LD50 assessment has a proper control (no iodine), but there is no control for the ddH2O trial or the LB broth trial. The data for the ddH2O trial should be removed from the manuscript. It is not applicable to aquaculture. No aquaculture organisms can survive in distilled water and no one uses it for culture. Soft water, i.e., with low hardness and alkalinity, is used, but that was not tested in this study. To test the effects of organic matter, I would recommend adding several cm of pond mud and pond water to aquaria; after settling, add bacteria, then iodine; sample at the various time intervals. This would be more representative of the actual organic load encountered in the pond.
2. Another concern in the application of the data is the potential to create bacteria strains that become more resistant to iodine at marginal concentrations or those that are protected in part by biofilms or organics in the mud. This is not addressed at all in your discussion, but should be mentioned as a concern to users of the results.
3. The authors imply that using aquaculture water for their research was a better way to evaluate the effectiveness of the iodine than using sterile water. The reason sterile water is used is to rule out effects of other bacteria, which cannot be done in this study. There was no confirmation of any kind noted in the paper that the bacteria observed on the agar plates were A. hydrophila. So the bacteria could have been any species that was in the water sample. This affects interpretation of the results, which indicated that the effective doses were for all culturable (heterotrophic) bacteria that can grow on LB media, not A. hydrophila per se. The discussion should be approached with this fact in mind.
4. The figures need to be redone to increase the font size and make them more readable.
5. Add a reference or recipe for the LB agar to the methods
6. Line 107: a sentence is needed describing how water samples were added to the bacteria pellet
7. L128: it is not clear how beads are used; please describe and reference as needed
8. L142: add duration data
9. Any tank aeration during the eel exposure trial?
10. Add details and conditions to the Methods section regarding the long-term monitoring of eels (up to 1 month)
11. L242: as mentioned under item 1, ddH2O has higher mortality because of the lack of Ca, Mg and the bicarbonate ions, not reduced organic matter. See Claude Boyd’s book on water quality in aquaculture ponds for more details and references on water quality. Deleting the ddH2O data will make this section unnecessary anyway.

---

## Round 0.3 · Minor Revisions

Your paper still needs some remaining minor changes. Please, review the entire text in order to avoid spelling mistakes and typos.

·

Basic reporting

Generally well written with appropriate references and introduction to the research problem.

Experimental design

no comment

Validity of the findings

data should provide additional information on the utility of iodine for disinfection and the effects of organic matter on its efficacy.

Additional comments

just a few editorial suggestions, given by line number:
L83, 'includes' not include
L87, 'uses' not use
L149, 'status' not statues
Add/move probit analysis to statistical analysis section
L184 change 'killed' to 'kill'
L231 delete 'in' so sentence reads '...was present...'

---

## Round 0.4 · accepted · Accept

After a carefull revision of your manuscript I am pleased to confirm that your paper has been accepted for publication in PeerJ.

Thank you for submitting your work to this journal.

#